# Dopamine Measurement Using Engineered CNT–CQD–Polymer Coatings on Pt Microelectrodes

**DOI:** 10.3390/s24061893

**Published:** 2024-03-15

**Authors:** Mahdieh Darroudi, Kevin A. White, Matthew A. Crocker, Brian N. Kim

**Affiliations:** Department of Bioengineering, Erik Johnsson School of Engineering & Computer Science, University of Texas at Dallas, Richardson, TX 75080, USA; mahdieh.darroudi@utdallas.edu (M.D.); kevin.white@utdallas.edu (K.A.W.);

**Keywords:** biosensor, neurotransmitter, dopamine, carbon quantum dots (CQDs), carbon nanotube (CNT), conductive polymers, fast-scan cyclic voltammetry (FSCV), PPy, PEDOT

## Abstract

This study aims to develop a microelectrode array-based neural probe that can record dopamine activity with high stability and sensitivity. To mimic the high stability of the gold standard method (carbon fiber electrodes), the microfabricated platinum microelectrode is coated with carbon-based nanomaterials. Carboxyl-functionalized multi-walled carbon nanotubes (COOH-MWCNTs) and carbon quantum dots (CQDs) were selected for this purpose, while a conductive polymer like poly (3-4-ethylene dioxythiophene) (PEDOT) or polypyrrole (PPy) serves as a stable interface between the platinum of the electrode and the carbon-based nanomaterials through a co-electrodeposition process. Based on our comparison between different conducting polymers and the addition of CQD, the CNT–CQD–PPy modified microelectrode outperforms its counterparts: CNT–CQD–PEDOT, CNT–PPy, CNT–PEDOT, and bare Pt microelectrode. The CNT–CQD–PPy modified microelectrode has a higher conductivity, stability, and sensitivity while achieving a remarkable limit of detection (LOD) of 35.20 ± 0.77 nM. Using fast-scan cyclic voltammetry (FSCV), these modified electrodes successfully measured dopamine’s redox peaks while exhibiting consistent and reliable responses over extensive use. This electrode modification not only paves the way for real-time, precise dopamine sensing using microfabricated electrodes but also offers a novel electrochemical sensor for in vivo studies of neural network dynamics and neurological disorders.

## 1. Introduction

Monitoring dopamine in the brain has played a critical role in studying dopaminergic neural circuits and neurological dysfunctions associated with the regulation of neurotransmitters [1,2,3]. Acquiring in vivo dopamine measurements requires methodologies that deliver real-time data with improved sensitivity and selectivity because of dopamine’s low extracellular concentration and potential measurement interference from other substances in the brain [4,5,6]. Conventional dopamine detection techniques such as microdialysis often struggle to provide real-time data due to slower response times [2,7,8]. Among various electrochemical methods of neurotransmitter detection, fast-scan cyclic voltammetry (FSCV) has priority due to its synergistic integration of high sensitivity, selectivity, and temporal acuity [1,9]. Through the redox mechanism, electroactive molecules such as dopamine can give electrons (oxidation) to or gain electrons (reduction) from the electrochemical sensing electrode at various voltage potentials using FSCV. The current signal detected by the electrode is directly proportional to both the concentration of analyte at the electrode as well as the number of electrons involved in the redox process. Additionally, electroactive molecules exhibit distinct voltammograms due to the differing potentials at which the molecules undergo oxidation or reduction. Through these mechanisms, the concentration of an electroactive molecule can be quantified based on the magnitude of current recorded. Also, the specific molecule undergoing the redox process can be identified based on its distinct voltammogram generated using FSCV [4,10]. Furthermore, FSCV has been a prominent electroanalytical technique offering remarkable temporal resolution in tracking rapid neurotransmitter changes in vivo [1,4,5]. Pioneered by Miller in the 1980s and subsequently popularized by Wightman, FSCV has evolved significantly over the years [11,12,13]. Background-subtracted FSCV has been a basis technique for differentiating and characterizing readily oxidizable biological compounds, notably dopamine [14,15]. Tuning the scan rate of the triangular waveform used in FSCV can optimize our differentiation of the electroactive species’ irreversible and quasi-reversible reaction peak potentials, allowing enhanced discrimination between interfering substances and mitigation of issues at the switching potential [4,10].

In FSCV, carbon-fiber microelectrodes (CFMEs) are the gold standard due to their high precision, minimal tissue disruption, as well as their excellent spatial resolution and sensitivity [16,17,18]. However, CFMEs are limited in their scalability. As such, integration into complex microelectrode arrays to provide spatial resolution is particularly challenging [19,20]. Multielectrode arrays through microfabrication engineering offer the advantage of high yields, batch production, and spatial accuracy. These arrays are instrumental in recording neurophysiological signals across various brain regions with exceptional clarity. Typically, microfabricated multielectrode arrays are constructed using metals like gold (Au) and platinum (Pt) on rigid silicon bases. Gold and platinum are commonly used due to their favorable microfabrication compatibility and robust electrochemical properties [21,22]. In microfabrication systems, microelectrodes can be engineered with precise geometry to optimize the detection of neurotransmitters, enable the monitoring of several analytes simultaneously, or allow for multiplexed detection in complex biological environments. Additionally, the ease of surface modification makes them ideal for spatially resolved measurements. However, metal electrodes exhibit inadequate sensitivity for dopamine, rendering them unsuitable for direct dopamine detection. Additionally, they demonstrate a propensity for performance degradation under the rigorous electrochemical conditions demanded by FSCV [19,20]. Persistent exposure to FSCV’s aggressive potential sweeps can lead to physical and chemical alterations of these metal surfaces, culminating in decreased sensor functionality. Consequently, the limitation underscores the need for advanced electrode materials that can withstand the rigorous conditions of FSCV without compromising the integrity and longevity of the neurochemical sensors.

Carbon is a desirable material for fabricating a dopamine sensor, so there is a pressing need to develop a scalable microfabrication technique to mass-produce carbon electrode arrays, whereby multiple devices can be patterned on a single wafer [22]. Recent advancements in neurochemical detection have markedly enhanced electrode technology for in vivo neurotransmitter monitoring. It was also demonstrated that arrays consisting of glassy carbon microelectrodes coated by CNT–PEDOT enables concurrent, multisite monitoring, which is crucial for electrophysiological studies [23,24]. Carbon nanomaterials are known for their inertness and sensitivity, providing a unique opportunity for research in electrochemistry [25]. The ideal scenario envisions using structured carbon materials such as aligned graphene sheets or carbon nanotubes (CNTs) with oriented edge planes for dopamine measurements [26,27]. Specifically, electrodes based on CNTs are employed in electrochemical biosensors owing to their high conductivity and rapid electron transfer [28]. These attributes make them highly suitable for in vivo applications, particularly for detecting rapid changes in the level of extracellular neurotransmitters, as they contain catalytic sites crucial for the oxidation of electroactive species [29,30]. Moreover, carbon quantum dots (CQDs) have recently gained prominence due to their remarkable properties, including low toxicity, biocompatibility, solubility, and applicability to a wide range of applications [31,32]. CQDs are becoming ubiquitous in sensor fabrication owing to their superior physiochemical and photoelectric characteristics, which stem from their unique structural properties, making them an excellent material for electrochemical sensing [33].

While carbon-based nanomaterials are not inherently compatible with traditional microfabrication techniques, electrode deposition of such nanomaterials onto gold or platinum can be used to produce a carbon-based microelectrode array. However, carbon nanomaterials cannot be electrodeposited. A novel approach must be developed to enable the microscopic patterning of carbon nanomaterials. Here, conducting polymers like poly(3-4-ethylene dioxythiophene) (PEDOT) or polypyrrole (PPy) play a pivotal role in providing an interface between CNTs and microelectrode surfaces [34]. They enhance both the adhesion and stability of CNTs and augment electron transfer efficiency [35], thereby improving the electrodes’ sensitivity and durability in biological contexts. By combining them with carbon nanomaterials, a hybrid nanocomposite with superior stability can be achieved [36].

Our study delves into the electrodeposition of carbon-based nanomaterials interlaced with conductive polymer matrixes onto platinum microelectrodes (100 × 30 μm), in order to enhance their stability and sensitivity, focusing on FSCV-based neurochemical sensing (Figure 1). This investigation is crucial for advancing our understanding of the electrochemical properties of platinum microelectrodes modified with CNT and their applicability in neurobiological contexts, potentially making an advancement in neurotransmitter dynamic analysis with improved spatial resolution. The novelty of our study lies in the unique integration of CNT–CQDs within conductive polymer matrices directly onto microelectrode arrays using electrodeposition, an approach not extensively studied for acute resolution applicability. This method aims to produce stable high-density carbon-based microelectrode arrays for in vivo neural applications. We aim to establish a reliable and efficient method to produce these arrays, potentially impacting the field of neural engineering.

## 2. Materials and Methods

### 2.1. Solution and Chemicals

Dopamine hydrochloride, EDOT, citric acid (Analytical Reagent grade, ≥99.5%), PBS (10×) pH 7.4, and Py were procured from Fisher Scientific (Fisher Scientific, Hampton, NH, USA). A stock solution of 1 mM of dopamine was prepared in PBS (1×, pH~7.4) and diluted subsequently to 10 μM with PBS (pH adjusted to 7.4). Dopamine concentrations used for electrochemical experiments are within the range of 100.0 nM to 1.0 μM, and the pH was carefully adjusted to 7.4.

### 2.2. Preparation of Carbonized Nanocomposite Modified Microelectrodes

For CQD synthesis, 2 g of citric acid was heated for approximately 3 h, maintained at a temperature of 220 °C, during which time it gradually changed color, becoming a homogeneous orange liquid [37]. Subsequently, an aqueous sodium hydroxide solution (10 mg/mL) was added to the liquid dropwise and stirred vigorously until the pH reached 6.5. The CQD solution was stored at 4 °C until use.

The probe containing Pt microelectrodes was constructed using microfabrication. First, a 4-inch silicon wafer with a 300-nm silicon nitride layer was purchased (UniversityWafer, Inc., South Boston, MA, USA). A Pt layer was patterned using photolithography and a lift-off process. Subsequently, a 200-nm layer of silicon oxide was deposited to serve as an insulation layer. This was followed by photolithography and plasma etching to create electrode openings (30 × 100 μm) (Figure 1). The electrodes then underwent low-pressure oxygen plasma treatment utilizing a Plasma-Flow PDC-32G (Harrick Plasma Inc., Ithaca, NY, USA) to remove residual organic contaminants and initiate surface hydroxylation, facilitating subsequent surface modifications. After plasma cleaning, the electrodes were rinsed with deionized (DI) water and dried with nitrogen.

### 2.3. Preparation of CNT–PEDOT and CNT–CQD–PEDOT Modified Microelectrodes

The Pt microelectrodes were exposed to a low-pressure plasma cleaner for 5 to 8 min, rinsed with DI, and dried over nitrogen. The fabrication of electropolymerized CNT–PEDOT and CNT–CQD–PEDOT modified Pt microelectrodes were carried out using an electrodeposition process. Initially, the COOH-MWCNTs were dispersed in PBS (pH = 7.4) and sonicated for 30 min to create a 1.5 mg/mL solution. As a next step, EDOT (0.02 M) or EDOT (0.02 M)–CQD (0.05 mM) was added into the CNT solution and then sonicated for another 30 min to ensure even dispersal. The Pt microelectrode (30 × 100 μm) was used as the working electrode. A silver/silver chloride (Ag|AgCl) wire was used as the counter and reference electrodes. Through constant current deposition, the CNT–CQD–PEDOT and CNT–PEDOT layers were fabricated on the electrode surface at a current of 2 nA/μm^2^ (vs. Ag|AgCl reference) with a deposition time of 1500 s and 1200 s, respectively (Figure 2d,f). Lastly, the carbonized microelectrodes were cleaned several times with DI water to remove residual EDOT and CQDs.

### 2.4. Preparation of CNT–PPy and CNT–CQD–PPy Modified Microelectrodes

The Pt microelectrodes were exposed to a low-pressure plasma cleaner for 5 to 8 min, rinsed with DI, and dried over nitrogen. Then, electrodeposition techniques were used to fabricate CNT–PPy and CNT–CQD–PPy on Pt microelectrode surfaces. Initially, COOH-MWCNT was dispersed in KCl (0.1 M) and sonicated for 30 min to create a 1.3 mg/mL solution. As a next step, pyrrole (0.02 M) or pyrrole (0.02 M)–CQD (0.05 mM) was added into the CNT solution and sonicated for another 30 min to ensure even dispersal. During the experiment, a Pt microelectrode (30 × 100 μm) served as the working electrode, with a silver/silver chloride (Ag|AgCl) wire as reference and counter electrodes. Using cyclic voltammetry (CV), a triangular potential was swept from −0.4 V to 1.4 V (vs. Ag|AgCl reference) at a scan rate of 50 mV/s for ten cycles, and the CNT–CQD–PPy and CNT–PPy layers were deposited on the Pt microelectrode surface. Lastly, the carbonized microelectrodes were cleaned several times with DI to remove residual PPy and CQDs (Figure 2c,e). It was found that increasing the number of cycles to 20 or beyond when CNTs were present resulted in higher resistivity, likely due to the formation of a thick CNT–PPy coating.

### 2.5. Electrochemical Analysis

FSCV and CV profiles of the bare microelectrodes and modified microelectrodes were obtained using a PalmSens4 (PalmSens, Utrecht, The Netherlands). Data collection and analysis were performed using PsTrace software (version 5.9.4515) and custom data acquisition hardware. A triangular waveform was applied to the electrode with a scanning potential from −0.5 to 1.4 V at a sweep rate of 300 V/s for FSCV of the electrodes with a silver/silver chloride wire serving as the reference electrode. Testing was conducted using a flow injection apparatus, utilizing a 4-port HPLC loop injector on a two-position air actuator. A PBS (1×) buffer and dopamine solution were conveyed through the flow cell at a 2 mL/min rate via a syringe pump. The response to the fast concentration changes were determined through background subtraction. Measurements were conducted before and after electrode modification to ensure proper process control.

## 3. Results

This study explores enhancing dopamine detection via FSCV using CNT and CNT–CQD coatings on Pt microelectrodes. Two distinct conductive polymers in conjunction with CNT, with or without the incorporation of CQDs, were produced (Figure 2) and characterized using a SEM (Figure 3), with the sensitivity and stability of the CNT–CQD–PPy and CNT–CQD–PEDOT modified Pt microelectrodes (further referred to as the CQD–Pt microelectrodes) examined by FSCV (Figure 4 and Figure 5).

### 3.1. Surface Characterization

The CQD–Pt microelectrodes were characterized using SEM (Figure 3). SEM images were acquired on a Zeiss ULTRA-55 FEG SEM microscope with a secondary electron detector operating at an accelerating voltage of 15 kV. All data is presented as mean ± the standard error of the mean. Due to the surface functionalization and ultrasound treatment, nanoparticles are not aggregated significantly (Figure 3c,g). SEM analysis is utilized to assess the structure and morphology of CNT–CQD–PEDOT and CNT–CQD–PPy. The SEM image– of the CNT–CQD–PEDOT layer indicate highly structured MWCNTs in the mesoporous structure of the polymer with uniform granular morphology. The porosity of the nanocomposites can be attributed to gas evolution during electropolymerization (Figure 3f,g) [38]. The surface morphology of CNT–CQD–PPy shows that the nanocomposite contains MWCNTs with a size below 50 nm. The nanocomposite contains non-agglomerated MWCNTs, confirming good dispersion of MWCNTs in PPy and the formation of a conductive network (Figure 3b,c). The porosity of the nanocomposite allows greater electrolyte access to the electroactive species due to increased surface area [9]. An EDAX analysis of CNT–CQD–PPy and CNT–CQD–PEDOT confirms successful coating of nanocomposite by elementary analysis of carbon (C), oxygen (O), sulfur (S), and nitrogen (N) (Figure 3d,h). The SEM and EDAX analyses confirm the successful fabrication of CNT–CQD–PEDOT and CNT–CQD–PPy layers.

### 3.2. Electrochemical Characterization

The inherent catalytic abilities of CNTs provide a strong foundation for facilitating the electrochemical reaction of dopamine. However, the integration of CQDs within these nanocomposites plays a significant role in augmenting the catalytic effect. The surface modification of the nanocomposites (CNT–CQD–PPy, CNT–CQD–PEDOT, CNT–PEDOT, and CNT–PPy) are characterized using CV in a solution of 10 mM [Fe(CN)_6_]^3−^/[Fe(CN)_6_]^4−^ (Figure 4a). This observation aligns with the CV data in which the bare electrode fails to display noticeable peaks, signifying that electrochemical oxidation and reduction of ferricyanide and ferrocyanide are not spontaneously initiated on the bare Pt microelectrode (Figure 4a). In the CVs of the modified microelectrode, current increases are attributed to surface modification with various carbonized nanocomposites. Specifically, the current values for CNT–CQD–PEDOT, CNT–CQD–PPy, CNT–PEDOT, CNT–PPy, and the Pt microelectrode are 75.06 ± 0.41 µA, 34.63 ± 0.56 µA, 19.79 ± 0.53 µA, 13.14 ± 0.37 µA, and 6.9 ± 0.24 µA, respectively. Therefore, the CV profiles of the CNT–PEDOT, CNT–CQD–PEDOT, CNT–PPy, and CNT–CQD–PPy modified microelectrodes show notably enhanced peak currents, indicating enhanced electron transfer kinetics.

Furthermore, the potential-peak separations between the E_anodic peaks_ and E_cathodic peaks_ of CNT–CQD–PEDOT (83 mV) and CNT–CQD–PPy (105 mV) are much smaller than those for CNT–PPy (230 mV), CNT–PEDOT (185 mV), and bare Pt microelectrodes (350 mv). The observed increase in peak current and decrease in peak separation in the electrochemical characterization of the CNT–CQD–PEDOT and CNT–CQD–PPy nanocomposites demonstrate the enhancement of catalytic properties of the dopamine redox reaction [39]. Figure 4a indicates improved ionic interchange between the electrolyte and the reactive substances due to surface modification, with the CNTs and CQDs contributing to the amplification of the electrochemical redox reaction. This analysis confirms that the CNT–CQD–PPy and CNT–CQD–PEDOT microelectrodes exhibit superior conductivity compared to their uncoated counterpart and nanocomposite layers without CQDs (Figure 4a).

FSCV was conducted in PBS (1×) with a pH of 7.4 for each modified microelectrode, as depicted in Figure 4b,c. The nanocomposites incorporating CQDs display a significant enhancement of their background currents compared with their non-CQD counterparts, suggesting that the addition of CQDs sharply alters the electrochemical profile of the microelectrodes.

FSCV remains the standard method for real-time neurotransmitter quantification. This study utilizes FSCV to assess the dopamine detection capabilities of various nanoelectrodes: CNT–PEDOT, CNT–PPy, CNT–CQD–PEDOT, CNT–CQD–PPy, and bare Pt. Our objective is to determine the impact of the nano-coated layers on the background current. Figure 4b,c display the FSCV data and background charging current for the bare electrode, and the coating layers of CNT–CQD–PEDOT and CNT–CQD–PPy. Notably, the CNT–PPy and CNT–PEDOT coated layers moderately elevate the background current compared to the bare Pt microelectrode. After integrating CQDs, there is a further increase in the background charging current.

Given that the background current is proportional to the surface area, it can be assumed that the CQDs, possessing small dimensions and a high surface area, significantly increase the overall surface area of the nanocomposite [40]. The increased surface area due to incorporating CQDs is evident in the recorded background current signals shown in Figure 4b,c. Thus, because a large surface area also increases the sensors’ sensitivity to electroactive neurochemicals for electrochemical detection using FSCV [4], we will focus on the performance of CNT–CQD–PEDOT and CNT–CQD–PPy for further analysis in Section 3.3 and Section 3.4.

### 3.3. Electrochemical Recordings of Dopamine

The background–subtracted FSCV of the microelectrodes coated with the PEDOT and PPy conductive polymers combined with CNTs and CQDs in 1-µM dopamine is shown in Figure 5a. The dopamine undergoes oxidation to dopamine-o-quinone at approximately 0.7 V, with reduction occurring at about −0.2 V. The average increase in background current signal of CNT–CQD–PEDOT and CNT–CQD–PPy modified Pt microelectrodes are shown in Figure 5b and demonstrate the layers’ significant effect on background current. The background current signal of the CNT–CQD–PPy layer is higher than that of the CNT–CQD–PEDOT layer; both the CNT–CQD–PEDOT and CNT–CQD–PPy modified Pt microelectrodes exhibit a profile higher than that of the bare Pt microelectrode. The CNT–CQD–PPy layer on the Pt microelectrode leads to an approximately 5-fold increase in background current, while the CNT–CQD–PEDOT increases roughly 4-fold (Figure 5b). When the CNT–CQD–PPy Pt microelectrode is used, the dopamine peak current experiences an approximately 1.5-fold increase, while the reduction peak current is enhanced approximately 1.4-fold relative to CNT–CQD–PEDOT (Figure 5c).

The conducting polymers affect both anodic and cathodic peaks (Figure 5c). The magnitude of the oxidation peaks is greater than that of the reduction peaks of CNT–CQD–PPy and CNT–CQD–PEDOT; this is attributed to dopamine’s more robust adsorption compared with dopamine-o-quinone. Incorporating CNTs and CQDs into PEDOT enhanced the oxidation peak current by roughly 2-fold, while CNT–CQD–PPy amplified the dopamine peak by 3-fold compared to CFMEs (if adjusted for the electrode area) [41]. Conversely, the reduction peaks of CNT–CQD–PPy and CNT–CQD–PEDOT experience a roughly 3-fold enhancement compared to the bare CFME microelectrode (Figure 5c) [41]. The CNT–CQD–PPy modified Pt microelectrode shows enhanced oxidation and reduction peak levels compared to a standard CFME and its PEDOT counterpart.

The correlation between scan rate (100–500 V/s) and anodic peak current for both the CNT–CQD–PPy and CNT–CQD–PEDOT have been investigated (Figure 5d,e). A linear relationship is observed between scan rate and anodic peak current. This relationship shows a slope of 0.8097 (R² = 0.9814) for CNT–CQD–PPy and 0.7566 (R^2^ = 0.9888) for CNT–CQD–PEDOT. In both the CNT–CQD–PPy and CNT–CQD–PEDOT layers, the anodic peak current is proportional to the scan rate, approaching a value close to 1, suggesting that the dopamine redox reaction is controlled by adsorption (Figure 5d,e) [4].

### 3.4. Dopamine Sensitivity

The dopamine sensitivity of carbon-based microelectrodes was tested using background-subtracted FSCV to measure the oxidative peak current against various dopamine concentrations for both the CNT–CQD–PPy and CNT–CQD–PEDOT layers (Figure 6). To assess the linear detection range, the modified microelectrodes were tested against various dopamine concentrations (0.1–1.0 μM) and were shown to exhibit linear responses (CNT–CQD–PPy with R^2^ = 0.9985 and CNT–CQD–PEDOT with R^2^ = 0.9718) (Figure 6). The sensitivity of CNT–CQD–PPy is 154 nA/μM, making it appropriate for measuring low concentrations of dopamine in the brain, while the sensitivity of CNT–CQD–PEDOT is 112 nA/μM.

The dopamine peak is distinctly visible for CNT–CQD–PPy at a concentration of 100 nM (Figure 6a). The calculated limit of detection (LOD) (S/N = 3) from the 100 nM data are 35.20 ± 0.77 nM for CNT–CQD–PPy and 40.06 ± 0.39 nM for CNT–CQD–PEDOT layers.

### 3.5. Stability of Engineered Microelectrodes

In FSCV, the electrode is continually scanned to ascertain its stability until a constant, drift-free response is observed. The stability analysis of the CNT–CQD–PPy and CNT–CQD–PEDOT layers using FSCV are displayed in Figure 7a,b. We intended to apply the FSCV waveform for a period of four hours using a flow cell instrument. During the four-hour test of the CNT–CQD–PPy layer, no significant change occurred in its background current (Figure 7a). In contrast, the background current of the CNT–CQD–PEDOT layer demonstrated a drifting pattern after the first hour. When observed with an optical microscope, it was apparent that the CNT–CQD–PEDOT layer was no longer adhered to the microelectrode surface, so the experiment was concluded at that time (Figure 7d). The CNT–CQD–PPy layer remained adhered to the microelectrode’s surface by the conclusion of the four-hour experiment period (Figure 7c).

An FSCV recording of dopamine was taken to ensure that the layers would not degrade and would maintain their reliability over time. The subtracted background results were compared with subsequent recordings after seven and fourteen days. The background-subtracted current for the CNT–CQD–PPy layer remained mostly unchanged, confirming the stability for a standard study (Figure 7e). Meanwhile, the CNT–CQD–PEDOT layer experienced both drift and a decrease in redox peaks (Figure 7f).

## 4. Discussion

The advantages of the FSCV technique over other electrochemical methods for real-time dopamine detection have been exhibited in Table 1. Although Differential Potential Voltammetry (DPV) methods using a microfabricated system provide high sensitivity, they cannot meet the high temporal resolution required for neurotransmitter measurements [39,42,43]. CNT–PEDOT modified glassy carbon multi-electrode arrays enable integrated multichannel measurements using Square Wave Voltammetry (SWV) for dopamine detection. However, the temporal resolution of this method is lacking [23]. The FSCV technique is the golden standard due to its high temporal resolution, selectivity, and sensitivity when studying neurotransmitter dynamics for in vivo studies. However, most studies on neurotransmitter measurement based on FSCV have extensively used CFMEs [3,44,45]. Providing spatial resolution without complex microelectrode arrays is particularly challenging for CFME. Designing electronics for microelectrode arrays that can be used for FSCV is demanding. Microfabrication engineering enables high yields and scalability for multielectrode arrays, which can accurately detect and record the neurotransmitters of neurophysiological signals.

While this is the first study involving direct coating of CNT–CQDs within a conductive polymer scaffold (PPy and PEDOT) stabilized on Pt microelectrodes (30 × 100 μm), previous studies have demonstrated that conductive polymers infused with MWCNTs enhance the electrochemical performance of microsensors and increase sensitivity when deposited on Pt electrodes [39,49].

SEM images and electrochemical data give insights into the carbonized layers and their impact on electrochemical signals. As depicted in Figure 2, both nanocomposites covered the whole microelectrode surface. While CNT–CQD–PEDOT forms a consistent layer exhibiting significant electrochemical properties, CNT–CQD–PPy displays greater sensitivity to dopamine, which is attributed to its influence on the faradaic current. The CNT–CQD–PEDOT layer and the CNT–CQD–PPy layer showed, respectively, an increase of four and five times the background charging current of a bare Pt microelectrode. This result implies that the surface modification of the electrodes influences their surface area, directly increasing their sensitivity [23,24]. The CNT–CQD–PEDOT electrode’s oxidation peak was two-fold larger while that of the CNT–CQD–PPy electrode was three-fold larger compared that of a similarly sized CFME [41]. Conversely, both nanocomposites underwent a three-fold increase in their reduction peak.

In the case of CNT–CQDs, it is suggested that oxygen functional groups amplify dopamine adsorption [3]. Techniques like oxygen plasma cleaning and laser treatment can provide more active sites on the edge plane to enhance sensitivity further. However, maintaining a balance in the amount of oxygen functionalized into the carbonized layer is essential to ensure both adsorption and conductivity. Integrating CNT–CQDs into PEDOT and PPy synergistically improves the electrical conductivities and charge transfer mechanisms.

However, after approximately one hour of repetitive FSCV cycling, the CNT–CQD–PEDOT layer experienced a significant degradation in its structural stability. This raises concerns about the nanocomposite’s long-term durability and reliability in practical applications. In comparison, the CNT–CQD–PPy layer remained in good condition after a period four times longer than the CNT–CQD–PEDOT survived.

These nanocomposites possess antifouling characteristics, which can be attributed to the zwitterionic properties of the negatively charged acid functionalized CNTs and CQDs. This aligns with findings reported in prior studies on various carbon-based nanomaterials; similar antifouling effects have been demonstrated for CNTs [24,50], Yarn CNTs [51], carbon nano horns [52], and graphene [53]. These studies exhibited that the negative charge on the COOH-CNTs and CQDs contributes to repelling biofouling agents, thereby enhancing the stability and durability of the electrode interface in biological environments [24]. Future research will explore further carbon nanocomposites in microelectrode arrays, improvements in sensitivity, and multisite as well as multi-neurotransmitter detection capabilities using FSCV, with the purpose of providing neurochemical insights and furthering the understanding of the brain’s electrochemical landscape.

## 5. Conclusions

We evaluated different nanocomposite layers with uniform carbonization on Pt microelectrodes for their sensitivity and stability. The electrodeposition of CNT–CQD–PPy emerged as the superior nanocomposite, resulting in a 3-fold increase in both anodic and cathodic peak currents compared to a CFME, while achieving an LOD of 35.20 ± 0. 77 nM. The CNT–CQD–PEDOT nanocomposite revealed approximately the same signal enhancement as CNT–CQD–PPy but showed relatively poor stability. In contrast, the CNT–CQD–PPy nanocomposite consistently demonstrated remarkable stability on the electrode surface. Because in vivo recording sessions typically last a few hours or longer, the added stability of CNT–CQD–PPy coated microelectrodes can help the wide use of microelectrode array-based neural probes in neurochemical detections. Overall, this study pioneers the direct application of CQDs alongside CNTs in polymer matrixes on the surface of Pt microelectrodes and highlights the effectiveness of this approach for FSCV dopamine measurement.

## Figures and Tables

**Figure 1 sensors-24-01893-f001:**
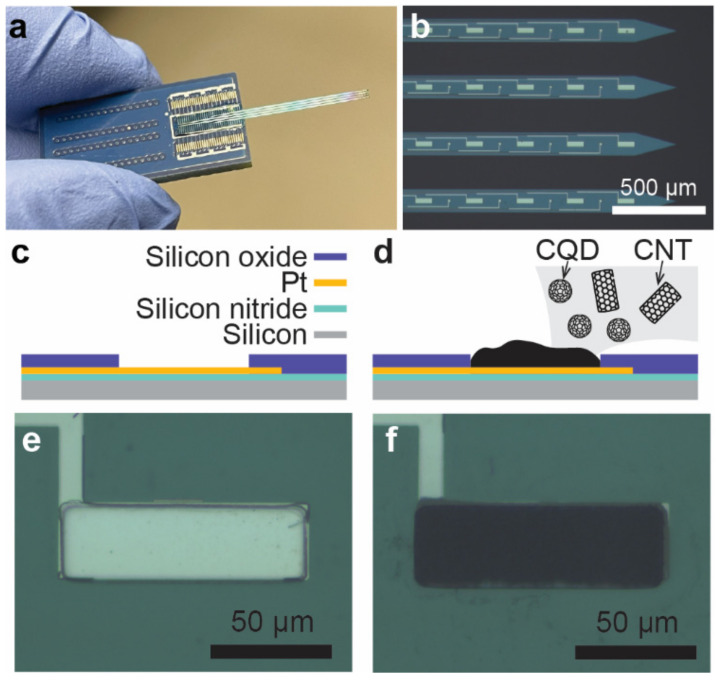
Carbonized Pt microelectrode fabrication for FSCV-based dopamine detection. Photographs of (**a**) the probe and (**b**) tip of the probe taken using a Zeiss Examiner; schematic views of (**c**) the microfabricated Pt microelectrode and (**d**) the carbonized Pt microelectrode, 30 × 100 μm; and micrographs of (**e**) the bare Pt microelectrode and (**f**) carbonized Pt microelectrode, 30 × 100 μm.

**Figure 2 sensors-24-01893-f002:**
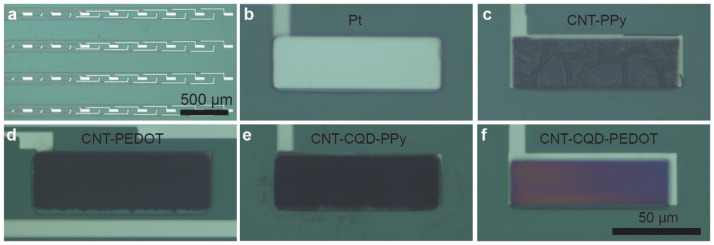
Electrodeposition of carbon-based nanomaterials on Pt microelectrodes. Photographs of the (**a**) entire microelectrode assembly (5×); (**b**) bare Pt microelectrode (100 × 30 μm) before nanocomposite deposition (50×); and (**c**) CNT–PPy, (**d**) CNT–PEDOT, (**e**) CNT–CQD–PPy, and (**f**) CNT–CQDs–PEDOT layers uniformly deposited on the Pt microelectrode (50×). These images were taken using a Zeiss Examiner.

**Figure 3 sensors-24-01893-f003:**
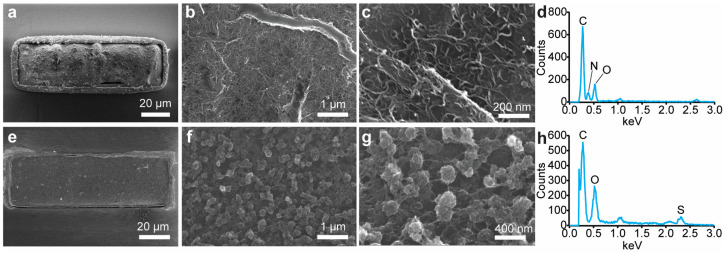
SEM images of CNT–CQD–PPy coated onto a Pt microelectrode (the scales are (**a**) 20 µm, (**b**) 1 µm, and (**c**) 200 nm) and (**d**) their EDAX analysis. SEM images of CNT–CQD–PEDOT coated onto a Pt microelectrode (the scales are (**e**) 20 µm, (**f**) 1 µm, and (**g**) 400 nm), and (**h**) their EDAX analysis.

**Figure 4 sensors-24-01893-f004:**
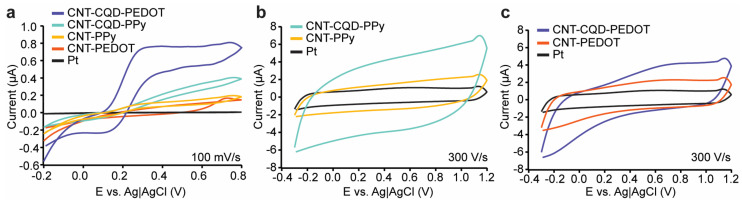
Electrochemical analysis of nanocomposite-coated Pt microelectrodes: (**a**) CV diagrams at a scan rate of 100 mV/s; (**b**) FSCV of CNT–PPy, CNT–CQD–PPy, and Pt bare microelectrodes at a scan rate of 300 V/s; (**c**) FSCV of CNT–PEDOT, CNT–CQD–PEDOT, and Pt bare microelectrodes at a scan rate of 300 V/s.

**Figure 5 sensors-24-01893-f005:**
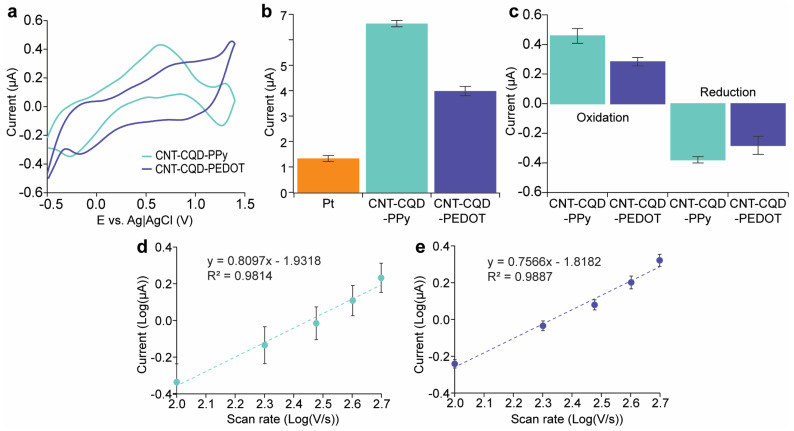
Comparative Analysis of the CNT–CQD–PPy and CNT–CQD–PEDOT layers on Pt microelectrodes: (**a**) background-subtracted FSCVs with 1 μM dopamine, (**b**) background charging current (average background charging current (*n* = 3)), (**c**) mean anodic and cathodic peak currents with 1 μM dopamine (*n* = 3), and the experimental scan rate performance for (**d**) CNT–CQD–PPy (*n* = 3) and (**e**) CNT–CQD–PEDOT (*n* = 3) at a scan rate of 300 V/s.

**Figure 6 sensors-24-01893-f006:**
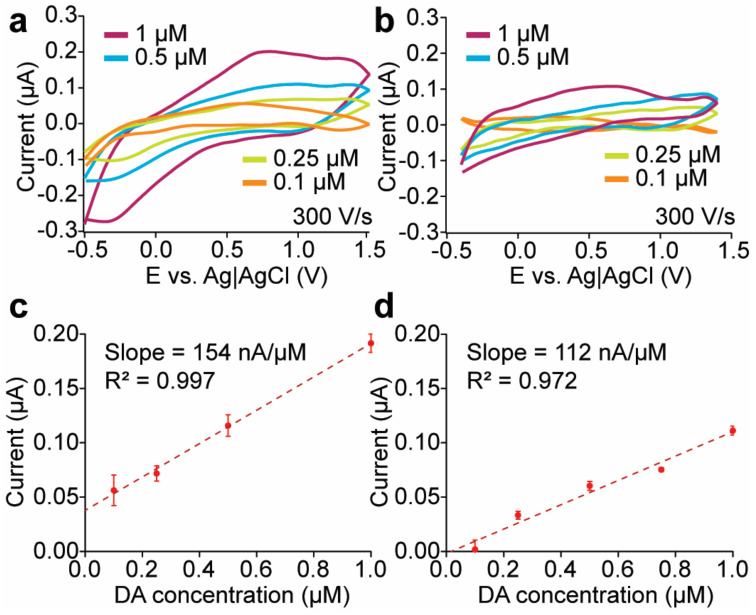
Concentration dependence analysis. The dopamine sensitivity for (**a**) the CNT–CQD–PPy layer and (**b**) the CNT–CQD–PEDOT layer. The relationship between the anodic peak current and dopamine concentration for (**c**) the CNT–CQD–PPy layer and (**d**) the CNT–CQD–PEDOT layer. Both plots demonstrate a linear response within the 100 nM to 1 μM dopamine concentration range (*n* = 3) at a scan rate of 300 V/s.

**Figure 7 sensors-24-01893-f007:**
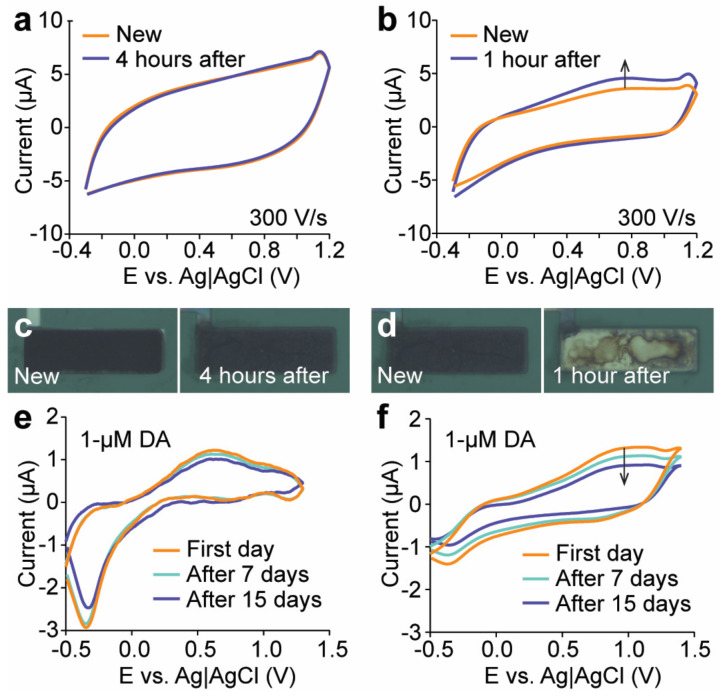
Evaluation of electrode stability. Stability testing in PBS (1X) for (**a**) CNT–CQD–PPy (over a four-hour period, *n* = 3) and (**b**) CNT–CQD–PEDOT (over a one-hour period, *n* = 3). Continuous waveform application and measurements were performed within potential windows of −0.3 to 1.2 V at a scan rate of 300 V/s. Every 30 min, the microelectrodes were observed using an optical microscope. Microscopic images of the electrode before and after the stability test for (**c**) CNT–CQD–PPy and (**d**) CNT–CQD–PEDOT. Weekly repeated background-subtracted FSCV stability tests for (**e**) CNT–CQD–PPy and (**f**) CNT–CQD–PEDOT. Measurements were obtained at 7-day intervals within potential windows of −0.5 to 1.5 V at a scan rate of 300 V/s (*n* = 3). The arrows indicate the direction of drift during the stability test.

**Table 1 sensors-24-01893-t001:** Comparison of major electrochemical techniques using carbonized electrodes for dopamine detection both in this work and with similarly reported sensors.

Ref.	Composite	Electrode	Method	Linear Range (µM)	LOD (nM)	Surface Area (µm^2^)	Sensitivity (nA/µM)
[42]	CNT–PPy	Au electrode (MEA)	DPV	0.005–10	0.14	-	0.5
[43]	GO–PEDOT	Au electrode		0.01–100	8	1256	0.87
[39]	CNT–PEDOT	Carbon paste electrode		0.1–20	20	1.25 × 10^7^	44
[46]	CNT–PEDOT	Silicon-based Microelectrode Arrays (MEA)	SWV	0.1–1	82	1200	108.3
[24]	GO–PEDOT	Glassy carbon (MEA)		0.01–1	56.2	1256	40.0
[38]	CNT–PEDOT	Carbon paste electrode	Amperometry	1.1–125	300	1.25 × 10^7^	0.25
[47]	rGO–PEDOT/Nafion	Au electrode		0.05–75	170	1256	9.9
[44]	CNT–PPy	CFME	FSCV	0.05–5	3.3	1278	11
[3]	GO			0.025–1	11	2238	41
[48]	CNT–Nafion			0.05–100	0.8	1278	1840
[45]	CNT–yarn			0.05–25	10.8	4240	2.5
[24]	GO–PEDOT	Glassy carbon		0.1–1	-	1256	400
**This work**	CNT–CQD–PEDOT	Pt electrode		0.1–1	40	3000	154
	CNT–CQD–PPy			0.1–1	35	3000	112

## Data Availability

All data are contained within the article.

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
