# Peer review of "Dopamine Measurement Using Engineered CNT–CQD–Polymer Coatings on Pt Microelectrodes"

_sensors, 2024, doi:10.3390/s24061893_

Round 1

Reviewer 1 Report

Comments and Suggestions for Authors

In this work, the electrochemical properties of platinum microelectrodes modified with CQD-CNT-Polymer are studied deeply, and an electrochemical sensor for real-time and accurate detection of dopamine is constructed. This work is interesting and inspiring. However, there are many mistakes and obscureness. Thus, I think that this paper would be modified carefully before publication in this journal.

My comments are summarized as below:

1. A variety of material have been developed for dopamine electrochemical sensing, please list a table showing the properties of this method compared to previous studies.

2. In Figure 3, the illustration is not clear, please improve it.

3. Please add the label in Figure 5a to descript the two curves clearly.

4. In Figure 6c, there is no corresponding current value when the dopamine concentration is 0.8 μM. Please explain it and the stability of the sensor.

5. Please provide some literature support for the conjecture in part 3.2.

6. It was written in the paper that "the prepared nanocomposite Pt microelectrode was characterized by SEM", but it was not fully reflected in the subsequent drawings.

7. “2.3. Preparation of CNT-PEDOT and CNT-CQDs-PEDOT modified microelectrodes” “2.3. Preparation of CNT-PPy and CNT-CQDs-PPy modified microelectrodes” “2.3. Electrochemical analysis” should be reorganized.

Comments on the Quality of English Language

Extensive editing of English language required.

Reviewer 2 Report

Comments and Suggestions for Authors

The manuscript titled "Dopamine Measurement using Engineered CQD-CNT-Polymer Coatings on Pt Microelectrodes" presents a novel approach for dopamine detection utilizing engineered coatings composed of carbon quantum dots (CQD), carbon nanotubes (CNT), and polymer on platinum (Pt) microelectrodes. The study investigates the efficacy of these coatings in enhancing the sensitivity and selectivity of dopamine detection, which is crucial for various applications including neuroscience research and medical diagnostics. The manuscript appears to offer valuable insights into the development of advanced sensing platforms for neurotransmitter detection. The manuscript can be recommended for publication after major changes and amendments.

1.     Line 114, which university wafer? The author should clarify it.

2.     In the manuscript the author didn’t mention whether they bought the platinum microelectrode or prepared it in his lab.

3.     Line 137, add the water or H2O after DI.

4.     Line 120, add the instrument name, by which the instrument images were taken e.g., confocal fluorescent microscopy, and SEM.

5.     Figure 2, line 183 which microscopy?

6.     Line 218, PBS concentration?

7.     The conclusion should be a separate part, it should not be with discussion.

8.     The manuscript didn’t perform a selectivity assay with another neurotransmitter.

9.     The author didn’t check the antifouling property.

10.  The sensing mechanism is not clear; they should mention the sketch or principle in the first section of the result.    

11.  The authors should discuss these two references in the introduction, results, and discussion which will strengthen the manuscript https://doi.org/10.3390/bios12070540,            https://doi.org/10.1039/D1AN00425E

Reviewer 3 Report

Comments and Suggestions for Authors

CNT-CQD-PPy and CNT-CQD-PEDOT coated layers on Pt Microelectrodes are used for dopamine determination using fact-scan cyclic voltammetry. The article provides interesting fundamental results but some details should be improved.

1. It is necessary to add in the introduction part with novelty of the used approaches. All components which are used for electrode modification are often used in sensor and biosensors, thus the novelty of the article should be more clearly indicated.

2. line 99. ,, buffer saline 1× (pH ~ 7.4),,. Is need to correct. What does ,, 1×,, mean?

3. There are 3 sections with 2.3 number.

4. Section ,,Electrochemical analysis,, should be divide, because Scanning Electron Microscopy isn’t electrochemical analysis.

5. Please, check the caption of Figure 3. Not all letters are deciphered. Is ,,um,, is nm or μm?

6. line 186. How was the spots averaging obtained with such a scale in Fig. 3? Please, add information how the confidence interval was determined.

7. on the fig. 3 d and h. there is a peak at about 1,0 keV? Why is there no comment, its value is comparable to the sulfur that was found.

8. line 214. Please, add error bars for current values.

9. Isn't the increase in background current when adding CNT, CQD, PEDOT or PPy an obvious thing? More conductive components increase the background current. Are there reported examples when adding conductive components decrease the background current?

10. Fig. 6. Please, add error bars for slopes.

11. fig. 6. It’s not entirely correct to name titration plots; you don’t find the equivalence point from them.

12. line 308. The same number of digits after point must be observed for the average and confidence interval. The 35.20 ± 0.077 should be correct.

13. For CV curves the scan rate should be added.

14. The analytical system was developed to monitor dopamine in the brain. Compare your linear dopamine range and the average range of concentrations in the brain?

15. Compare the analytical parameters of your device with other reported sensor or biosensors for dopamine detection.

16. Finally, add the practical significance of the results: how could the system potentially be used to monitor dopamine in the brain? What will the sample be for analysis?

17. In the article, there is no information about gold standard method (Carbon fiber electrode) only in abstract.

In general, the work is important in the context of online monitoring of main analysts in the brain. Such work may be allowed to be published, but with corrections and suggested improvements.

Round 2

Reviewer 1 Report

Comments and Suggestions for Authors

 Accept 

Reviewer 2 Report

Comments and Suggestions for Authors

The authors addressed all the comments and have made significant modifications to the manuscript.